# Mutual Interactions between Brain States and Alzheimer’s Disease Pathology: A Focus on Gamma and Slow Oscillations

**DOI:** 10.3390/biology10080707

**Published:** 2021-07-23

**Authors:** Nicole Byron, Anna Semenova, Shuzo Sakata

**Affiliations:** Strathclyde Institute of Pharmacy and Biomedical Sciences, University of Strathclyde, 161 Cathedral Street, Glasgow G4 0RE, UK; nicole.byron.2015@uni.strath.ac.uk (N.B.); anna.semenova.2020@uni.strath.ac.uk (A.S.)

**Keywords:** dementia, Alzheimer’s disease, neuromodulation, neural oscillations, optogenetics

## Abstract

**Simple Summary:**

Electrical activity in the brain dynamically changes throughout the day. Abnormalities in brain activity have been associated with various brain disorders, including Alzheimer’s disease (AD). While brain disorders stem from complex pathological processes, resulting in abnormalities in neural activity and cognitive deficits, recent studies have demonstrated that controlling brain activity can modify disease pathologies as well as cognitive functions. In particular, studies in mouse models of AD have provided promising results regarding the amelioration of AD pathology by invasive and non-invasive brain stimulations. In this review article, by focusing on AD, we provide an overview of this emerging field. We summarise how brain activity changes in humans and mouse models, and how different artificial manipulations of brain activity can modify AD pathology. Although further investigations are essential, this research direction will provide insight into non-pharmacological intervention strategies for dementia.

**Abstract:**

Brain state varies from moment to moment. While brain state can be defined by ongoing neuronal population activity, such as neuronal oscillations, this is tightly coupled with certain behavioural or vigilant states. In recent decades, abnormalities in brain state have been recognised as biomarkers of various brain diseases and disorders. Intriguingly, accumulating evidence also demonstrates mutual interactions between brain states and disease pathologies: while abnormalities in brain state arise during disease progression, manipulations of brain state can modify disease pathology, suggesting a therapeutic potential. In this review, by focusing on Alzheimer’s disease (AD), the most common form of dementia, we provide an overview of how brain states change in AD patients and mouse models, and how controlling brain states can modify AD pathology. Specifically, we summarise the relationship between AD and changes in gamma and slow oscillations. As pathological changes in these oscillations correlate with AD pathology, manipulations of either gamma or slow oscillations can modify AD pathology in mouse models. We argue that neuromodulation approaches to target brain states are a promising non-pharmacological intervention for neurodegenerative diseases.

## 1. Introduction

The brain is never at rest. The activity state of the brain, called the brain state, varies from moment to moment. While brain state can be defined as a collective action of the neural population at a given moment, it spans over multiple spatiotemporal scales (Figure 1) [1,2,3]. Hans Berger first described the 8–12 Hz rhythm, called alpha waves, in a human scalp electroencephalogram (EEG) recording [4]. Since then, intensive research has discovered a wide range of activity patterns or brain states (Figure 1A). For example, gamma (30–90 Hz) oscillations are a fast activity state and appear locally, compared to slower frequency oscillations. Gamma oscillations are related to various cognitive functions, such as attention, conscious perception and memory [5,6,7,8,9,10,11,12]. The sleep–wake cycle can be considered as slower state changes and consists of multiple neural events and oscillations (Figure 1). Rapid eye movement (REM) sleep is characterised by theta oscillations and ponto-geniculo-occipital (PGO) waves, whereas non-REM (NREM) sleep is characterised by slow oscillations, sleep spindles, and sharp wave-ripples [13,14,15,16,17,18,19,20,21] (Figure 1B). These sleep-related neural events or oscillations have also been implicated in various homeostatic and cognitive functions, including waste clearance [22,23] and memory consolidation [24,25,26,27,28,29].

Given the prominence of these neural oscillations and events, it is not surprising that consistent associations can be seen between various brain disorders and abnormalities in neural oscillations or brain states [30,31,32,33,34]. For example, abnormalities in gamma oscillations have been recognised as a neurophysiological marker for various neuropsychiatric disorders and neurodegenerative diseases, such as schizophrenia [34], autism spectrum disorder (ASD) [35,36], depression [37,38], and Alzheimer’s disease (AD) [31]: more specifically, a reduction in sensory-evoked gamma power can be seen in schizophrenia and ASD patients, whereas varied changes in gamma oscillations have been reported in depressive disorders and AD patients [34,36,38,39,40,41]. Additionally, abnormalities in sleep patterns and sleep-related oscillations have been linked with depression [42], schizophrenia [43,44], addiction [45] and AD [46,47,48,49]. 

Although it remains to be determined how abnormalities in brain states can be causally linked to disease pathogenesis, an emerging approach, called “neuromodulation”, aims to alter neural activity to modify disease state [50]. For example, while deep brain stimulation (DBS) is an invasive approach, chronically implanting a depth electrode into the patient brain to electrically stimulate a target brain region, it can alleviate the symptoms of Parkinson’s disease [51,52] and has also been examined in treatment-resistant depression [53,54], obsessive-compulsive disorder [55,56] and AD [57,58]. In addition to invasive treatment, non-invasive neuromodulation approaches, such as transcranial magnetic stimulation (TMS), have been explored in various brain disorders, such as schizophrenia [59,60,61], depression [62], addiction [63,64], and AD [65,66,67]. Despite many clinical trials being conducted, it remains unclear how neuromodulation approaches can act on neural circuits to result in cellular and molecular responses that modify disease state. To tackle this challenge, preclinical studies in animal models could offer insight into better neuromodulation approaches. Thus, it is crucial to understand how brain state is regulated, how brain state is changed during disease pathogenesis, and how neuromodulation approaches can alter neural activity, resulting in a modification of disease state. 

In this literature review, we focus on AD, the most common form of dementia. Although AD is one of the most intensively studied neurodegenerative diseases [68,69,70,71,72,73,74,75,76,77,78], intervention and treatment options remain limited. In AD, amyloid plaques and tauopathy are major pathological hallmarks, with other pathological features including inflammation and lipid metabolism [68,70,71,73,74,75,77,78,79,80]. Although molecules associated with these features are promising targets for pharmacological treatments [81,82,83,84,85], neuromodulation-based interventions are now being considered, given the multifaceted pathologies of AD. 

Abnormalities in EEG patterns have been recognised since as early as the 1930s [30]. Since then, EEG abnormalities have been described in terms of the following three features [31]: (1) slower neural oscillations, (2) decreased complexity of EEG, and (3) reduced degrees of functional connectivity. Hence, these hallmarks of EEG abnormalities can be recognised as either a biomarker of or target for neuromodulation-based intervention. Indeed, accumulating evidence indicates that neuromodulation approaches have the potential to modify Alzheimer’s disease states [47,86]. In particular, targeting gamma oscillations and slow (<1 Hz) oscillations has provided encouraging results in AD mouse models [87,88,89,90,91,92]. Since these oscillations have been well characterised with respect to their induction mechanisms, gamma and slow oscillations would make good targets for neuromodulation-based treatment. 

In this review, we first summarise the mechanisms of these two neural oscillations, gamma and slow, followed by a brief overview of the relationship between these oscillations and AD in both human patients and mouse models. Then, we review recent animal studies that examined the effect of invasive and non-invasive neuromodulation approaches on AD pathology. Finally, we discuss future directions in this field. Readers may also refer to other recent reviews relevant to this field [33,46,47,48,49,67,86,93]. 

## 2. Gamma Oscillations and AD

Jasper and his colleague first described gamma waves [94]. The investigation of gamma (30–90 Hz) oscillations has gained popularity following a series of studies by Freeman [95,96] and by Singer and his colleagues [9]. Gamma oscillations have been observed across many brain regions, not just in the neocortex, but also in the entorhinal cortex [97,98,99], amygdala [100], hippocampus [101,102,103], striatum [104,105], olfactory bulb [106,107], basal forebrain [108,109] and developing thalamus [110]. They have been associated with various cognitive functions, including attentional selection [8,111], working memory [12,112], perceptual binding [6,9], and memory encoding [111,113]. Abnormalities in gamma oscillations have links to various neuropsychiatric disorders [34] and neurodegenerative diseases [32]. Here, after describing the induction mechanisms of gamma oscillations, we summarise the relationship between gamma oscillations and AD in both humans and mouse models. Then, we discuss emerging therapeutic approaches based on gamma oscillations. 

### 2.1. Mechanisms of Gamma Oscillations

The induction mechanisms of gamma oscillations in cortical circuits have been investigated by a wide range of approaches, including computational models [5,103,114,115,116,117], brain slice experiments [118,119] and in vivo optogenetic experiments [120,121]. Computational studies have suggested several mechanisms that are potentially involved in generating gamma oscillations (Figure 2) [5,114,122]. In the interneuron gamma (ING) mechanism (Figure 2A), mutual inhibition between GABAergic neurons is sufficient to generate gamma oscillations. Two distinct regimes can be considered: in the high-firing, noise-free condition, individual GABAergic neurons elicit spikes at around 40 Hz. Mutual inhibition via GABA_A_ receptors quickly leads to synchronous firing [103]. In the more realistic, noisy condition, individual GABAergic neurons fire sparsely and stochastically. When the inhibitory feedback is strong enough, gamma oscillations arise. Thus, gamma oscillations are an emerging property of the mutual inhibition network. In both conditions, GABA_A_ receptor-mediated inhibition plays a role in the generation of gamma oscillations in the absence of excitatory inputs [103,117]. 

In the pyramidal-interneuron gamma (PING) mechanism (Figure 2B), the alternation between the fast excitation and delayed feedback inhibition can generate gamma oscillations [5,107,114,122,123]. The fast excitation is mediated by AMPA receptors, whereas the feedback inhibition is mediated by GABA_A_ receptors. The third, simple mechanism is the inheritance of gamma rhythm from upstream areas (Figure 2C) [124]. In this mechanism, the downstream network can reliably and precisely respond to rhythmic inputs from their upstream. In addition to these, other mechanisms can also be taken into consideration, such as neuromodulators [125,126] and pace-making chattering cells [127].

Experimentally, optogenetic activation of cortical parvalbumin-positive (PV+) GABAergic neurons is sufficient to produce gamma oscillations, whereas αCaMKII+ neurons (pyramidal neurons) cannot entrain optogenetic stimulation at the gamma-frequency range [120,121]. Although these results apparently support the ING mechanism, the determination of a precise mechanism is complicated. Although the computational models described above predict how the depolarization of excitatory or inhibitory neurons could affect the oscillations, they do not predict how each network configuration could respond to *periodic* optogenetic stimulations. Rather, models incorporating physiological data suggested that these optogenetic experiments cannot conclusively distinguish between the ING and PING mechanisms, since the PING model can explain the experimental observations [114]. Thus, further studies with different stimulation protocols are required to determine the mechanisms of gamma oscillations. As gamma oscillations may have therapeutic potential for neurodegenerative diseases, it is important to investigate whether different induction mechanisms of gamma oscillations could lead to distinct molecular and cellular responses and what types of induction mechanisms have beneficial or detrimental effects on AD pathology.

### 2.2. Gamma Oscillations and AD in Humans

While gamma oscillations in human subjects have been assessed by either EEG or magnetoencephalography (MEG), the relationship between AD pathology and changes in gamma oscillations is inconclusive: a number of studies reported a reduction in gamma power or coherence across cortical regions in AD patients [40,128,129], whereas some reported opposite results [39,41,130,131,132]. This inconsistency may stem from various experimental parameters. For example, gamma oscillations were assessed in an eye-closed, resting state condition [40,41,131,133,134] or during sensory stimulus presentation [39,41,129,130,132]. As expected, cortical regions which showed significant effects also varied depending on conditions and studies. Additionally, while many studies compared gamma oscillations between AD patients and healthy subjects, several studies also compared AD and mild-cognitive impairment (MCI) patients [41,130,131,134]. A consistent approach, recruiting a large number of subjects, would be ideal to address this issue. 

A recent study of >300 individuals provides valuable insight into changes in gamma oscillations during AD pathogenesis [135]. This study revealed the inverted U-shape relationship between amyloid depositions and gamma power (Figure 3): as the amyloid deposition reaches a supra-threshold level, gamma power increases. On the other hand, as the amyloid deposition increases further to an enhanced pathological state, gamma power decreases. These results imply a compensatory mechanism at an early phase of AD pathogenesis, which may be overwhelmed by a higher amyloid load, leading to a breakdown of neural circuits [135,136,137,138]. These results may also be reconciled with the contradictory observations mentioned above, as changes in gamma oscillations may vary depending on the stage of pathogenesis. In the future, it is important to correlate changes in gamma oscillations with AD pathology in large-scale clinical studies. In addition, it is crucial to find out whether animal models can replicate this inverted U-shape relationship to investigate the underlying mechanisms at the molecular, cellular and neural circuit levels.

### 2.3. Gamma Oscillations and AD in Mouse Models

A reduction in gamma power is consistently observed in various mouse models, including APP-PS1 [139], J20 [140,141], 5xFAD [89], CRND8 [142,143], APOE4 [144,145] and tau models [146] (Table 1). Multiple brain regions have been investigated, such as the hippocampus [89,143,145], entorhinal cortex [139] and prefrontal cortex [140,141]. In the hippocampus, abnormalities in the coupling of gamma oscillations with sharp wave-ripples or theta oscillations have been consistently observed [89,142,145]. While these results imply that the overexpression of amyloid-β impairs hippocampal ensembles, amyloid precursor protein (APP) also plays a critical role in theta-gamma coupling in the hippocampus, as mice lacking APP exhibit a reduction in theta-gamma coupling without statistically significant changes in gamma and theta power [147]. 

Additionally, an association between deficits in PV+ neurons and abnormal gamma oscillations has been shown [140,141]. More specifically, this is caused by a reduction in voltage-gated sodium channel subunit Nav1.1 expression in PV+ neurons of J20 mice, with experimental studies illustrating that genetic modifications to increase Nav1.1 expression lead to a restoration of gamma oscillations and a beneficial effect on cognitive decline [141]. Given the mechanisms of gamma oscillations described above (Figure 2), it is important to examine how the deficits in PV+ neurons can affect activity in pyramidal cells and interactions between PV+ and pyramidal neurons.

Intriguingly, a subset of interneurons (such as PV+, somatostatin-positive, and cholecystokinin-positive GABAergic neurons) are vulnerable to amyloid pathology [140,141,148], whereas pyramidal cells are more vulnerable to tauopathy [149]. These results suggest selective vulnerability depending on AD pathogenesis and pathologies. As efforts have been made to comprehensively characterise molecular mechanisms of such selective vulnerability in both humans and mouse models [149,150,151,152,153], deficits at the neural circuit level will also become clear in coming years.

Compared to human studies, the following aspects have been less explored in mouse models: firstly, although several studies have investigated multiple age points to show modifications in gamma oscillations during AD pathogenesis [136,145,154,155], none of them have reported the inverted U-shape change, that is, a transient increase in gamma power at an early phase of AD pathogenesis, as reported in a human clinical study (Figure 3). A longitudinal assessment of mouse models correlating with amyloid burden and other pathological features may address this issue. Secondly, commonly used mouse models are familial AD models. Thus, the relation to late-onset AD (LOAD) remains unclear. A recent effort to develop LOAD mouse models [156] may bridge the gap between human and animal studies. Additionally, the effect of tauopathy in gamma oscillations needs to be explored further. Finally, the electrophysiological approach is markedly different between human and mouse studies. For example, very few studies in mice have assessed sensory-evoked, task event-related or sensory steady-state responses. Additionally, cortex-wide gamma coherence has not been assessed in mice, in contrast to human EEG and MEG studies. Filling these methodological gaps will be crucial in the future. 

### 2.4. Neuromodulation of Gamma Oscillations for AD

As summarised above, it is clear that a reduction in gamma power is associated with AD pathology, at least in mouse models. Leveraging this knowledge, various invasive and non-invasive neuromodulation approaches have been adopted to modify AD pathology (Table 2) [88,89,90,91,157,158,159,160]. For example, Tsai and her colleagues have elegantly demonstrated that both invasive and non-invasive gamma stimulations can ameliorate AD pathology [86,89,90,91]: optogenetic induction of gamma oscillations in the hippocampus can reduce amyloid load by activating microglia [89]. More surprisingly, non-invasive 40 Hz sensory stimulation (either auditory or visual) has similar effects [89,90]. The same approach can also reduce tau phosphorylation and seeding in the T301S model [89,90]. These effects are associated with modifications to microglia-associated transcripts, as well as synaptic signaling and plasticity-related proteins [89,91]. Although the effect of this non-invasive approach remains to be confirmed in humans, multisensory 40 Hz stimulation can affect wider brain regions, including hippocampal areas, as well as sensory cortices [90]. Another group showed that optogenetic stimulation of PV+ neurons in the medial septum can induce gamma oscillations in the hippocampus, resulting in an improvement in cognitive function [88]. Although it remains to be determined whether this approach could also reduce amyloid load in the hippocampus, these studies have illustrated the potential for certain induction mechanisms of gamma oscillations to modify AD pathology in a beneficial manner. However, further investigation of these is required before its potential as a non-pharmaceutical therapy is considered. 

Interestingly, a recent alternative optogenetic approach to induce cortical gamma oscillations showed opposing effects on AD pathology [157]: although optogenetic activation of basal forebrain PV+ neurons could induce cortical gamma oscillations, amyloid load increased in the medial prefrontal cortex and septum. As basal forebrain PV+ neurons preferentially innervate cortical GABAergic neurons [161], the optogenetic activation of basal forebrain PV+ neurons could suppress cortical PV+ neurons, rather than activating them. Thus, the induction mechanism of cortical gamma oscillations in this study differs from that of Iaccarino et al. (2016) [89]. These results suggest that the beneficial effect of gamma oscillations may depend on the induction method, rather than the frequency of local field potentials itself. As there are multiple mechanisms to induce gamma oscillations (Figure 2), it is important to investigate how different approaches can activate different components of neural circuits as well as non-neuronal cells. This type of effort will refine this therapeutic option. As several parameters (duration, frequency, age, etc.) must be explored, real-time monitoring of AD pathology in vivo [162] will accelerate this field.

Regarding clinical applications, since a current major limitation is that most studies have focused on amyloid pathology (Table 2), it is important to investigate how the induction of gamma oscillations affects other pathological features, especially tauopathy [89,90]. In addition, because changes in gamma oscillations in humans can vary depending on the stage of AD pathogenesis [135], it is also critical to determine whether this neuromodulation approach could be beneficial even for patients who exhibit higher gamma power. Again, developing and examining better animal models will benefit this exciting research direction.

## 3. Slow Oscillations and AD

Slow (<1 Hz) oscillations are another well-characterised type of neural oscillation, since the series of landmark studies by Steriade and colleagues [20,163,164]. Slow oscillations are comprised of cycles of global silence (DOWN state) and synchronous firing (UP state) across neuronal populations [2,163,165,166,167]. When they appear during NREM sleep and under anaesthesia, they can predominantly be observed in the cerebral cortex [2,20,166,167,168,169,170], but also in other brain regions, including the thalamus [171,172], thalamic reticular nucleus [173], hippocampus [170], striatum [174,175], brainstem [176] and claustrum [177]. Slow oscillations play a role in sleep-dependent memory consolidation [26,29,178]. 

The sleep–wake cycle regulates the concentration of amyloid-β and tau in the cerebrospinal fluid (CSF) and interstitial fluid (ISF), with a higher level of amyloid-β and tau occurring due to prolonged wakefulness or sleep deprivation [179,180]. Slow oscillations are also linked to the activity of the glymphatic system, a highly organised CSF transport system, to clear protein waste products including amyloid-β [23]. Indeed, abnormalities in slow oscillations have been associated with AD [47,49]. Thus, these results suggest a close relationship between the glymphatic system degradation, sleep disturbance and disease progression in dementias [181]. 

Here, we summarise how slow oscillations are generated and how the reduction in slow wave activity correlates with AD pathology in human patients and mouse models. Finally, we discuss a therapeutic opportunity based on slow oscillations. Although slow oscillations are closely related to sleep, especially NREM sleep, we focus primarily on the oscillation itself and slow wave activity. Readers may refer to recent comprehensive reviews on sleep and AD elsewhere [46,47,48,49,182,183]. Readers can also refer to an up-to-date review of the detailed mechanisms of slow oscillations [184]. Although covering the detailed molecular mechanism is beyond the scope of this review article, transcriptomic and synaptic phosphorylation profiles related to sleep–wake cycles have recently been characterised [185,186].

Regarding terminologies, slow oscillations refer to oscillations at less than 1 Hz, whereas delta oscillations refer to oscillations at 1–4 Hz. However, delta oscillations are often described as 0.5–4 Hz oscillations in literature; hence, they may include slow oscillations. Slow wave activity (SWA) typically refers to spectral power around the 0.5–4 Hz range. SWA is closely associated with sleep homeostasis [187]: it increases proportionally to time spent awake and peaks in slow-wave sleep, whereas it decreases as sleep propensity is reduced. 

### 3.1. Mechanisms of Slow Oscillations

Earlier studies showed that slow oscillations can be generated in isolated cortical gyrus [188], a cortical slab [169] and cortical slice [189], suggesting that cortical circuits are sufficient for the generation of slow oscillations. Subsequent studies have consistently demonstrated that recurrent excitation of layer (L) 5 pyramidal cells is a source of slow oscillations [166,168,189,190]. This notion has been confirmed by computational studies, in which UP and DOWN states can be reproduced by models of neural populations with recurrent excitation and slow adaptation (e.g., activity-dependent K^+^ current or synaptic depression) (Figure 4A) [191,192,193,194,195]. 

Multiple receptors and ion channels contribute to shaping UP and DOWN states. For example, both NMDA and non-NMDA receptors are involved in the excitatory drive of UP states [189]. While both excitatory and inhibitory neurons are active during UP states (see below for more details), GABA_A_ receptors play a critical role in UP state duration [196]. The termination mechanism of UP states remains to be fully determined, as various processes have been proposed (for a review, see [184]).

Although recurrent excitation of L5 pyramidal cells plays a dominant role in slow oscillations, accumulating evidence has demonstrated a complex picture: while recurrent excitatory activity during UP states is balanced by inhibition [197], two major GABAergic cell classes, PV+ and somatostatin-positive neurons, regulate the transitions of UP and DOWN states [198]. A recent study also showed that deep-layer neurogliaform cells contribute to slow oscillations by preferentially firing during DOWN states [199]. These experimental results may favour the computational models, which implement active contributions from inhibitory neurons to the UP–DOWN dynamics (Figure 4B) [200,201].

With respect to subcortical areas, thalamic neurons play a critical role in the full manifestation of slow oscillations via T-type calcium channels in thalamocortical cells [171,172]. Thalamic neurons drive PV+ neurons during DOWN states [202]. PV+ neurons can be also activated by claustral neurons to induce DOWN states across cortical regions [177]. Moreover, it has been suggested that astrocytes play a role in slow oscillations and NREM sleep [203,204,205,206,207]. Thus, the exact mechanisms of slow oscillations remain to be fully determined [208]. While the detailed biophysical models of cortical columns [209,210] could provide valuable insight into the mechanisms of slow oscillations, implementing subcortical inputs and the non-neuronal components are still challenging.

Although slow oscillations can arise from various cortical areas as “slow waves”, they often start from the lateral and medial frontal cortical regions and propagate as travelling waves to posterior cortical areas in the human brain [211,212,213]. While cortex-wide spontaneous activity can be examined by various means, correlating different signals (e.g., electrical, hemodynamic, intracellular calcium signals) is still an open issue; this would help gain insight into the mechanisms of cortex-wide slow waves.

### 3.2. Slow Oscillations and AD in Humans

Sleep disturbance is a common symptom of AD pathogenesis, with sleep fragmentation, increased nocturnal activity and excessive daytime napping contributing to the disruptions to daily life [46,214,215,216,217,218,219,220]. Thus, it is not surprising to see the robust association between abnormalities in slow-wave sleep and AD pathology in humans [46,221,222,223,224,225,226]. Additionally, earlier studies in the 1980s and 1990s demonstrated an association between abnormalities in slow wave activity and AD pathology, including cognitive functions [227,228,229]. Specifically, studies show that impairments in slow-wave sleep are associated with impaired cognition. In recent decades, it has become evident that these associations are underpinned by structural changes and AD pathology in the brain: age-related prefrontal atrophy is associated with reduced slow-wave activity during NREM sleep [230]. Additionally, a bi-directional relationship between slow-wave sleep and AD pathology exists, as slow-wave activity during NREM sleep decreases as amyloid-β deposition and tau accumulation increase [231,232]. The reduction in slow-wave activity is also associated with the impairment in sleep-dependent memory consolidation [231]. Thus, changes in slow oscillations are a robust biomarker of AD pathogenesis in humans although underlying cellular and circuit mechanisms remain unclear. 

### 3.3. Slow Oscillations and AD in Mouse Models

The sleep-wake cycle has been examined across different pathological stages in various mouse models, including 3xTg-AD [233,234], APP/PS1 [234,235], Tg2576 [234,236], P301S Tau [237], rTG4510 [238], PLB1_Triple_ [239], and PLB2tau [240] models (Table 3). Slow (<1 Hz) oscillations have been analysed together with delta oscillations (1–4 Hz), which can be used to determine NREM sleep. In several AD mouse models, NREM sleep is reduced and fragmented [233,234,237,238], which implies that changes also occur in the patterns of slow oscillations. It has been suggested that the reduction in GABAergic tone impairs long-range synchronous firing in an amyloid mouse model [92]. Intriguingly, P301S Tau model exhibited the inverted U-shape profile at the delta frequency, meaning that delta power increases at an early disease stage, whereas it decreases at a later stage [237]. Longitudinal studies in AD mouse models may provide valuable insight into the mechanisms of age-related changes in slow oscillations. 

### 3.4. Neuromodulation of Slow Oscillations for AD

Pharmacological and optogenetic intervention approaches can modify abnormalities in slow oscillations, hence the AD disease state in mouse models (Table 4) [87,92,241,242]. For example, a breakdown of long-range coherence of slow oscillations in an AD mouse model can be rescued by enhancing GABAergic inhibition with a GABA_A_ receptor agonist [92]. This is consistent with the notion that aberrant somatic GABAergic tone plays a critical role in the hyperactivity of cortical neurons [140,243]. Kastanenka and his colleagues demonstrated frequency-specific effects of optogenetically induced cortical slow oscillations on AD pathology [87,241]: slow-wave-specific 0.6 Hz optogenetic stimulation of αCaMKII+ neurons in the anterior cortical area can reduce amyloid-β and increase GABA_A_ and GABA_B_ receptor expression. On the other hand, 1.2 Hz stimulation, a slight offset from a slow-wave-specific frequency band, shows an opposing effect without altering GABA_A_ and GABA_B_ receptor expression. These results imply that increasing inhibitory tone may play a role in reducing amyloid burden. One potential caveat is that, because increased and decreased firing rate can also modify AD pathology [244,245,246,247], optogenetic stimulation at higher frequencies may also induce a higher firing rate, leading to the promotion of amyloid deposition. Future studies are needed to determine whether and how the temporal structure of neural population activity, rather than simple firing rates, can modify AD pathology. Additionally, although abnormalities in slow and delta oscillations have been reported in tau models (Table 3), it remains to be explored whether artificial manipulations of slow oscillations can modify tauopathy as well as other pathological features.

Nevertheless, these studies suggest that pharmacological and non-pharmacological interventions for slow oscillations have therapeutic potential for AD. Indeed, accumulating evidence shows bidirectional relationships between sleep and AD pathogenesis [48,248]. It is important to investigate whether artificially enhanced slow oscillations can also trigger other non-neuronal events, such as glymphatic waste clearance, which can be seen in natural slow-wave sleep and even under anaesthesia [23,249]. As discussed above, because multiple components contribute to slow oscillations, various approaches could be explored to modify these and, hence, AD pathology. It is by taking advantage of these various approaches for the neuromodulation of slow oscillations that evidence for potential different modes of action on AD pathology will be unearthed. 

## 4. Conclusions and Future Directions

By focusing on gamma and slow oscillations, we summarised how these oscillations can change during AD pathogenesis in both humans and mouse models. We also reviewed emerging invasive and non-invasive neuromodulation approaches to modify AD pathology in mouse models based on gamma and slow oscillations. Although further studies are essential to uncover the underlying mechanisms before clinical applications, these neuromodulation-based interventions are promising frontiers for AD and beyond. 

To further explore this emerging field, the following four areas are important for investigation. Firstly, it is essential to comprehensively characterise electrophysiological biomarkers in AD animal models, with respect to neural oscillations. As well as offering potential biomarkers for early diagnosis, this will aid in the understanding of neural oscillations in AD and abnormalities throughout its progression. As we discussed above, although reduced gamma power has consistently been reported in mouse models (Table 1), the available evidence in human patients is conflicting [39,40,41,128,129,130,131,132]. The gap between mouse models and humans may be due to discrepancies in methodologies between studies, a lack of proper longitudinal studies in mouse models, or the limitations of animal models [250]. Since better mouse models which reflect human pathological features are under development [156], it will become important to conduct detailed in vivo electrophysiological investigations correlating with various molecular and cellular AD pathologies—not just amyloid pathology, but also tauopathy and other pathological features.

Secondly, it is worth exploring other brain states, not just gamma and slow oscillations, because a wide range of neural oscillations or events have been studied in general neuroscience [1,13,33,251]. With respect to sleep, the reduction in REM sleep duration is an early biomarker during AD pathogenesis [221,227,228,252]. During REM sleep, hippocampal theta rhythms and ponto-geniculo-occipital (PGO) waves are prominent electrophysiological markers [14,21,253,254]. Intriguingly, PGO waves and theta rhythms are temporally coupled across animal models [21,255,256,257]. Therefore, it is interesting to investigate how this functional coupling between several sleep-related neural events is affected during AD pathogenesis. PGO waves are originated from mesopontine cholinergic areas [17,19,258] and the neurodegeneration of mesopontine cholinergic neurons has been associated with AD [259,260], as well as Lewy body dementia [261]. Additionally, abnormalities in sleep spindles and sharp wave-ripples during NREM sleep have also been associated with AD [16,49,252,262]. Therefore, these sleep-related neural oscillations may be alternative targets for non-pharmacological interventions.

Thirdly, the means of neuromodulation needs to be explored further. Although optogenetic approaches can achieve cell type-specific manipulations with a high spatiotemporal resolution, non-invasive approaches are ideal for clinical applications. A chemogenetic approach, along with a detailed characterization of brain state, could be a promising direction, since a recent study demonstrated that the chemogenetic attenuation of hyperactivity in the entorhinal cortex can ameliorate AD pathology, including the spread of pathological tau [245]. At present, repetitive transcranial magnetic stimulation (rTMS) and transcranial direct or alternating current stimulation (tDCS/ACS) have shown promising results [65,66]. In addition to these brain stimulation methods, sensory stimulation is also an attractive approach to modulate gamma oscillations across brain regions [86,89,90]. Several neural oscillations or neural events during sleep are also known to be induced or modulated by sensory inputs [263,264,265]: for example, PGO waves during REM sleep can be triggered by sounds [265]. In addition, slow oscillations can be modulated by sounds to promote memory consolidation [263]. Thus, neuromodulations based on auditory stimulus during sleep may be an attractive option.

Finally, the most important research direction is to uncover the mechanisms of how the manipulation of neural oscillations can modify pathological features across multiple spatial levels, from molecular to neural circuit levels. Supposing that neurons are a key driver for a certain neural oscillation, how can subsequent molecular responses trigger non-neuronal events, such as microglial and astrocytic activation and the modification of neurovascular coupling? In addition, neural oscillations are typically induced by multiple neural circuit motifs [5,208,266]. A fundamental issue is to uncover the direct link among electrophysiological signatures, cell-type-specific neuronal activity, non-neuronal activity and molecular responses. Given the complexity of such interactions over multiple spatiotemporal scales, computational approaches will play a crucial role in better understanding the effects of neuromodulation approaches on AD pathology. Thus, we predict that integrative, systems-level approaches will become increasingly important in the coming years. 

In conclusion, bi-directional relationships between AD pathology and brain states have become evident. While gamma oscillations and slow oscillations are promising targets, many issues remain to be explored. For future clinical applications, it is crucial to establish a causal relationship between AD pathology and neuromodulations at various levels, from molecules to neural circuits.

## Figures and Tables

**Figure 1 biology-10-00707-f001:**
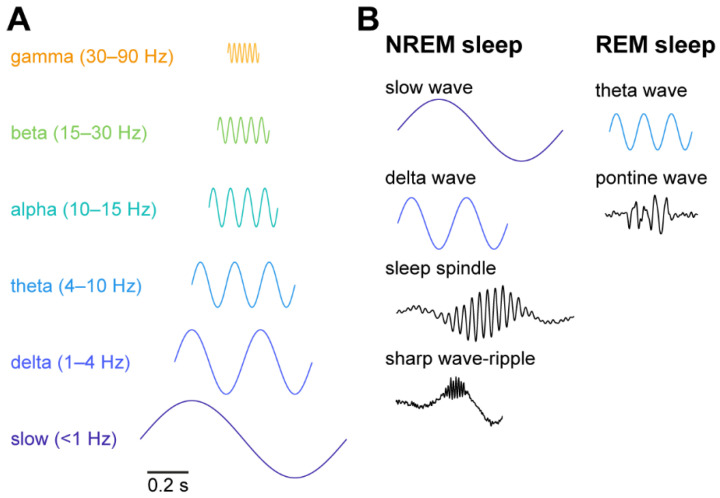
Neural oscillations and events across brain states. (**A**) Neural oscillations and their frequency band. (**B**) Characteristic neural oscillations and events during sleep states.

**Figure 2 biology-10-00707-f002:**
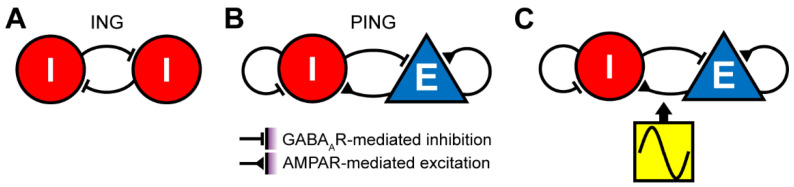
Mechanisms of gamma oscillations. (**A**) Interneuron gamma (ING) mechanism. Gamma oscillations arise from mutually connected GABAergic neurons. Two regimes can be considered: in one regime, each interneuron fires rhythmically with a frequency determined by the kinetics of the GABAergic feedback (~40 Hz). In the second regime, although each interneuron sparsely and stochastically fires at an average rate *below* 40 Hz, recurrent inhibitory interactions lead to gamma oscillations. (**B**) Pyramidal-interneuron network gamma (PING) mechanism. Pyramidal cells first activate interneurons via AMPA receptors (AMPARs). This leads to recurrent inhibition via GABA_A_ receptors (GABA_A_Rs), resulting in rhythmic firing of excitatory and inhibitory populations at the gamma range. (**C**) Gamma oscillations are inherited by oscillatory activity from upstream areas.

**Figure 3 biology-10-00707-f003:**
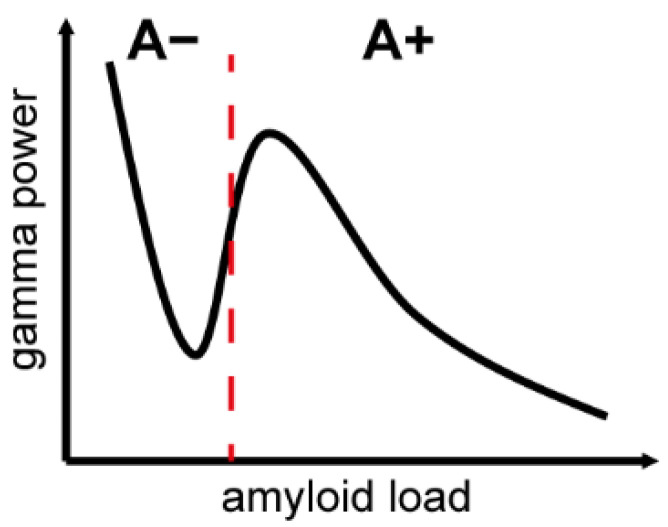
Inverted U-shape relationship between amyloid load and gamma power. This inverted U-shape relationship is evident in the neurodegeneration-positive group assessed by ^18^F-FDG PET. A+ and A−, amyloid-positive and -negative groups based on ^18^F-florbetapir PET, respectively. Please note that although these two groups were dichotomized with a threshold (red dotted line), amyloid load is a continuous value. Modified from Gaubert et al. (2019).

**Figure 4 biology-10-00707-f004:**
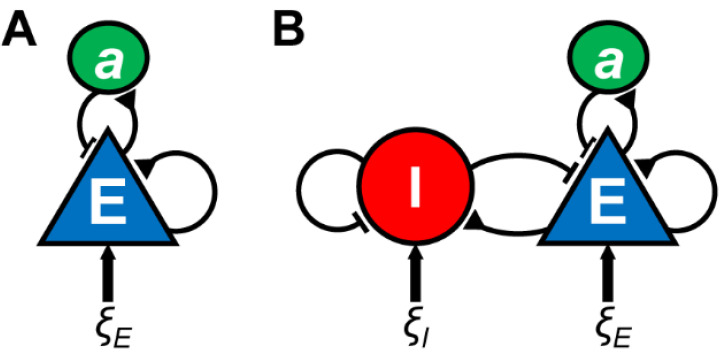
Models of slow oscillations. Two types of neural population (rate) models for UP and DOWN dynamics are illustrated. (**A**) A neural population model with recurrent excitation, slow adaptive process *a* and noisy fluctuations *ξ_E_*. (**B**) A neural population model with recurrent excitation and inhibition, slow adaptive process and independent noisy inputs. The physiological implementation of the adaptive process can be activity-dependent K^+^ current or synaptic depression.

**Table 1 biology-10-00707-t001:** Gamma oscillations in AD mouse models.

Mouse Model	Age (Months)	Sex	Preparation	Frequency Band (Hz)	Changes in γ Oscillations	Reference
APP/PS1	4–5	NA	EC slices	20–60	Reduced γ frequency in LEC No effect in MEC	[139]
J20	4–7	M/F	in vivo cEEG	20–80	Reduced γ power	[140]
	7–8	M/F	in vivo cEEG	30–90	Reduced γ power	[141]
5xFAD	3	M	in vivo LFP in CA1	20–50	Reduced γ power during SWRs	[89]
TgCRND8	1	NA	HC slices	θ: 3–12 low γ: 25–85 high γ: 120–250	No change in γ power Disrupted θ–γ coupling	[142]
	1	M	in vivo HC LFP	low γ: 25–45 high γ: 60–100	Reduced γ power	[143]
APOE4	5–17	F	in vivo HC LFP	30–50	Reduced γ power	[144]
	4–5	F	in vivo HC LFP	30–50	Reduced γ power during SWRs	[145]
3R tau overexpression	7	M	HC slices	50–90	Reduced γ power and peak frequency	[146]

cEEG, cortical EEG. EC, entorhinal cortex. HC, hippocampus. LFP, local field potential. LEC, lateral EC. MEC, medial EC. SWR, sharp wave-ripple.

**Table 2 biology-10-00707-t002:** Summary of invasive and non-invasive neuromodulation of gamma oscillations in AD mouse models.

	Induction Method	Stimulation Protocol	Duration	Model	Sex	Age (Months)	Modulated AD Phenotype	Reference
**Invasive**	Optogenetic	1 ms pulses, 40 Hz, CA1	1 h	5xFAD::PV-Cre, AAV5-EF1α-DIO-ChR2-eYFP	M	3	Reduced Aβ Reduced inflammation	[89]
		12 ms pulses, 40 Hz, Medial Septum	10 min	PVJ20, AAVdj-EF1α-DIO-ChETA-eYFP	M/F	NA	Improved spatial memory	[88]
		40 Hz, Basal Forebrain	1 h/d for 3 days	5xFAD::PV-Cre::Ai32	M/F	4–6	Increased Aβ	[157]
**Non-Invasive**	Visual	12.5 ms on, 12.5 ms off, 40 Hz flicker	1 h/day for 7 days	5xFAD	M	6	Reduced Aβ	[89]
				APP/PS1	M/F	5	Reduced Aβ	
				TauP301S	M	4	Reduced tauopathy	
		40 Hz flicker	1 h/day for 30 days	APP/PS1	F	8	Reduced Aβ Reduced tauopathy Increased sleep regulation	[160]
		12.5 ms on, 12.5 ms off, 40 Hz flicker	1 h/day for 22 days	TauP301S	M	7.5–8	Reduced neuronal damage Reduced inflammation Reduced tauopathy Improved spatial memory	[91]
			1 h/day for 6 weeks	CK-p25	M/F	6–9	Reduced neuronal damage Reduced inflammation Improved spatial memory	
	Auditory	1 ms 10 kHz tones, 40 Hz, 60 dB	1 h/day for 7 days	5xFAD	NA	6	Reduced Aβ Reduced inflammation Improved memory	[90]
				APP/PS1	NA	6–9	Reduced Aβ Reduced inflammation	
				TauP301S	NA	2	Reduced tauopathy	
	Combined Auditory and Visual	10 s on, 10 s off	1 h/day for 7 days	5xFAD	NA	6	Reduced Aβ Reduced inflammation	
	Visual and Exercise	40 Hz light flicker and 30–50 min exercise	Daily, 6 days a week for 12 weeks	3xTg	M	12–15	Reduced Aβ Reduced tauopathy Reduced neuronal damage Improved spatial memory	[159]
	Transcranial Focused Ultrasound	400 μs pulses, 5 s on 5 s off, 40 Hz, Hippocampus	1 h/day for 5 days	5xFAD	M	6	Increased microglia/Aβ Co-localisation	[158]

**Table 3 biology-10-00707-t003:** Slow and delta oscillations in AD mouse models.

Mouse Model	Age (Months)	Sex	Frequency Band (Hz)	Changes in Oscillations	Reference
3xTg-AD	7, 20	M/F	<1	Increased frequency at 7 months Decreased frequency at 20 months More irregular at 20 months	[233]
3xTg-AD	18	M/F	0.1–4	No change	[234]
APP/PS1	8–10	M/F	0.1–4	Decreased power during NREM	
Tg2576	12	M/F	0.1–4	Decreased power during W	
Tg2576	2, 6, 12	NA	0.5–4	Decreased power during NREM at 6–12 months	[236]
APP/PS1	3, 6, 9	NA	>1	Shorter NREM at 9 months	[235]
P301S	3–12	M	1–4	Increased power during NREM at 6–9 months Decreased power during W and NREM at 11 months	[237]
rTg4510	5–10	M	0.1–4	Decreased power during NREM from 6 months	[238]
PLB1triple	5–21	M/F	0.5–5	Decreased power during REM at 9 months Decreased power during W at 21 months	[239]
PLB2tau	6	F	1.5–5	Increased power during REM Decreased power during NREM	[240]

REM, rapid eye movement sleep. NREM, non-REM sleep. W, wakefulness.

**Table 4 biology-10-00707-t004:** Summary of invasive and non-invasive neuromodulation of slow oscillations in AD mouse models.

	Induction Method	Protocol	Duration	Model	Sex	Age (Months)	Modulated AD Phenotype	Reference
**Invasive**	Optogenetic	400 ms pulses, 0.6 Hz, Anterior Cortex	24 h/day for 1 month	APP/PS1, AAV5-CamKIIα-hChR2(H134R)-mCherry	M/F	4–7	Reduced Aβ Reduced calcium overload Restored GABA levels	[87]
		400 ms pulses, 1.2 Hz, Anterior Cortex	24 h/day for 1 month	APP/PS1, AAV5-CamKIIα-hChR2(H134R)-mCherry	M/F	3–9	Increased Aβ Increased calcium overload Decreased spine densityNo change in GABA levels	[241]
**Non-Invasive**	BACE Inhibitor (oral)	Administration of 0.25 g/kg NB-360 in food pellets	6 weeks ad lib	APP23xPS45	F	6–8	Reduced Aβ Reduced calcium overload Improved spatial memory	[242]
	GABA-A Agonist (i.p.)	Administration of 0.05 mg/kg clonazepam	Once/day for 5 days	APP23xPS45	M/F	6–8	Improved spatial memory	[92]

## Data Availability

All data are included in this published article as figures and tables.

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
