# Peer review of "Mutual Interactions between Brain States and Alzheimer’s Disease Pathology: A Focus on Gamma and Slow Oscillations"

_biology, 2021, doi:10.3390/biology10080707_

Round 1

Reviewer 1 Report

This current review examines the mechanisms by which gamma & slow wave oscillations are generated in the brain and assesses how these oscillations may be affected by Alzheimer’s disease pathology. Furthermore, the authors investigate how the emerging field of neuromodulation to enhance these oscillations may be applied to increase waste clearance or ameliorated AD pathology.

In general, this review is well written and informative.

However, the authors focus heavily on amyloid pathology, and since this review is examining interactions between brain states and Alzheimer’s disease pathology, this reviewer feels the authors may want to include more information on the role of other pathologies, especially Tau, in AD. A number of important papers on this have been missed.

Also, a more critical view on the amyloid hypothesis is needed, and the issues related to animal models of AD (failed reproducibility etc) need to be included.

It is necessary to list animal models with a clearer critical analyses as to their genetic design, pathology, face value, conflicting publications etc. Also, technical approaches to measure gamma rhythms have varied hugely, the impact of this is not sufficiently considered.

Specific points

Introduction:

  • The sentence ‘While gamma oscillations are a fast…’ on lines 43-45 is confusing, since the authors use brain/activity/vigilance/sleep state interchangeably, it can be hard to follow.
  • [Refs 63-67] on line 78 are too amyloid focused and don’t fully encompass the current understanding of AD pathology.
  • [Ref 29] and [Ref 3], there are multiple typos here (I appreciate this is in German but still….please correct).

Uber das Elektrenkephalogramm des Menschen. Archiv fur Psychiatrie md Nervenkrankheiten 1929, 87, 527-570.

Uber das Elektrenkephalogramm des Menschen. Dritte Mitteilung. Archiv fur Psychiapie ind Nervenbaizkheifen

If the authors are unable to read the German original articles, it may be better to cite papers published in English, see examples in https://dl.uswr.ac.ir/bitstream/Hannan/32444/1/9781472469441.pdf

Gamma Oscillations and AD

  • Sentence on lines 96-97 is odd, not sure gamma oscillations themselves can gain popularity, more like the investigation of gamma oscillations is getting more popular.
  • [Ref 30] on line 104 should actually be [Ref 31]?
  • Figure 2: the authors could label GABA & AMPA receptors to make their point even clearer (same for fig4)
  • Should ‘Magnet encephalography’ be one word – magnetoencephalography on line 158?
  • As before, the authors very much focus on amyloid deposition but tauopathy may also be contributing (differentially) to changes in neural oscillations. Of note, the role of fibrillar amyloid  / plaques is very controversial, and soluble species are more likely to be of disease relevance. This aspect has been ignored and all models are listed as being the same.

Slow oscillations and AD

  • The authors may want to include information on waking slow wave activity.
  • Figure 4: font size on lines 275-276 needs to be adjusted.
  • The authors could add references for Tg4510 mice and PLB2-Tau mice, as there have been good longitudinal studies in these models.

Conclusions and Future directions

  • Again, overall nicely written, clear and concise, but a more critical conclusion on amyloid vs other pathologies (Tau, ApoE etc) and mentioning of contradictory animal model data is needed here as well. It is not quite correct that all animal model data are consistent.
  • ‘increasing’ on line 417 should be ‘increasingly’.

Author Response

This current review examines the mechanisms by which gamma & slow wave oscillations are generated in the brain and assesses how these oscillations may be affected by Alzheimer’s disease pathology. Furthermore, the authors investigate how the emerging field of neuromodulation to enhance these oscillations may be applied to increase waste clearance or ameliorated AD pathology.

In general, this review is well written and informative.

Thank you for your constructive feedback. Our point-by-point responses are as follows.

However, the authors focus heavily on amyloid pathology, and since this review is examining interactions between brain states and Alzheimer’s disease pathology, this reviewer feels the authors may want to include more information on the role of other pathologies, especially Tau, in AD. A number of important papers on this have been missed.

Agree. To address this issue, we have made the following amendments:

Firstly, in line 82, we have added several key papers to cover a wide range of pathological features in AD, including a highly cited review on tauopathy (Lee et al., Ann Rev Neurosci 2001) and others.

Secondly, we have added descriptions about cell type-specific vulnerability to tauopathy in a mouse model. See lines 227-230.

Thirdly, in lines 258-260, we have added descriptions about how the induction of gamma oscillations can modify tau phosphorylation and other features.

Finally, in lines 393-394 as well as new Table 3, we have added studies in tau models by adding two citations.

Also, a more critical view on the amyloid hypothesis is needed, and the issues related to animal models of AD (failed reproducibility etc) need to be included.

Thank you for pointing out this critical point. We have made the following amendments:

Firstly, in lines 234-248, we have elaborated a critical analysis on discrepancies between human and animal studies and on the limitation of mouse models.

Secondly, in lines 284-287, 425-427 and 453-460, we have elaborated a critical analysis on a current limitation of mouse studies.

It is necessary to list animal models with a clearer critical analyses as to their genetic design, pathology, face value, conflicting publications etc.

We have added two new tables to summarise pathophysiology in various mouse models, with respect to gamma (Table 1) and slow oscillations (Table 3).

Also, technical approaches to measure gamma rhythms have varied hugely, the impact of this is not sufficiently considered.

We have elaborated our critical analysis on this issue by specifying varied approaches to measure gamma oscillations. Please see lines 175-183 and 234-248 and Table 1.

Introduction:

The sentence ‘While gamma oscillations are a fast…’ on lines 43-45 is confusing, since the authors use brain/activity/vigilance/sleep state interchangeably, it can be hard to follow.

We have clarified this. Please see lines 41-50.

[Refs 63-67] on line 78 are too amyloid focused and don’t fully encompass the current understanding of AD pathology.

As mentioned above, we have added citations to cover a wide range of pathological features in AD. Please see line 82 and references 68-78.

[Ref 29] and [Ref 3], there are multiple typos here (I appreciate this is in German but still….please correct).

Uber das Elektrenkephalogramm des Menschen. Archiv fur Psychiatrie md Nervenkrankheiten 1929, 87, 527-570.

Uber das Elektrenkephalogramm des Menschen. Dritte Mitteilung. Archiv fur Psychiapie ind Nervenbaizkheifen

If the authors are unable to read the German original articles, it may be better to cite papers published in English, see examples in https://dl.uswr.ac.ir/bitstream/Hannan/32444/1/9781472469441.pdf

Thank you. We have corrected the citations.  Please see Refs 4 and 30.

Gamma Oscillations and AD

Sentence on lines 96-97 is odd, not sure gamma oscillations themselves can gain popularity, more like the investigation of gamma oscillations is getting more popular.

Corrected. Please see lines 107-108.

[Ref 30] on line 104 should actually be [Ref 31]?

We have updated it. Please see line 116.

Figure 2: the authors could label GABA & AMPA receptors to make their point even clearer (same for fig4)

We have updated Figure 2.

Regarding Fig. 4, because NMDA receptors and other components may be involved, we have intentionally unspecified the connection types. Thus, it remains the same as the original one.

Should ‘Magnet encephalography’ be one word – magnetoencephalography on line 158?

Corrected. Please see line 172.

As before, the authors very much focus on amyloid deposition but tauopathy may also be contributing (differentially) to changes in neural oscillations. Of note, the role of fibrillar amyloid  / plaques is very controversial, and soluble species are more likely to be of disease relevance. This aspect has been ignored and all models are listed as being the same.

We have elaborated our critical analysis on this issue. Please see lines 234-248, 258-260 and 284-287, and Table 1.

Slow oscillations and AD

The authors may want to include information on waking slow wave activity.

We have clarified related terminology including slow wave activity by mentioning slow wave activity during wakefulness. Please see lines 323-329.

Figure 4: font size on lines 275-276 needs to be adjusted.

Corrected.

The authors could add references for Tg4510 mice and PLB2-Tau mice, as there have been good longitudinal studies in these models.

Thank you for the suggestions. We have added these studies. Please see lines 393-394 and Table 3.

Conclusions and Future directions

Again, overall nicely written, clear and concise, but a more critical conclusion on amyloid vs other pathologies (Tau, ApoE etc) and mentioning of contradictory animal model data is needed here as well. It is not quite correct that all animal model data are consistent.

We have elaborated our discussion. Please see lines 453-460.

‘increasing’ on line 417 should be ‘increasingly’.

Corrected. Please see line 501.

Reviewer 2 Report

The article reviews recent promising approaches to assessing non-pharmacological intervention strategies for dementia. The authors focus on Alzheimer's disease (AD), providing an overview of how brain states can influence AD pathology in patients and mouse models by neuromodulation of brain states. Specifically, the relationship between AD and changes in gamma and slow oscillations is summarized. The topic is exciting and is being intensively investigated. Besides, the manuscript presents promising studies on the potential manipulations of either gamma or slow oscillations on AD pathology in mouse models. This review article has used sufficient and appropriate scientific reports, discussing and contrasting previously published experiments on the subject. It is a fascinating topic, the conclusions are in concordance with the reviewed material, and the figures and table are adequate.
The review is exhaustive and is formally well thought out and argued with the referenced bibliography, so its publication is desirable.

Author Response

The article reviews recent promising approaches to assessing non-pharmacological intervention strategies for dementia. The authors focus on Alzheimer's disease (AD), providing an overview of how brain states can influence AD pathology in patients and mouse models by neuromodulation of brain states. Specifically, the relationship between AD and changes in gamma and slow oscillations is summarized. The topic is exciting and is being intensively investigated. Besides, the manuscript presents promising studies on the potential manipulations of either gamma or slow oscillations on AD pathology in mouse models. This review article has used sufficient and appropriate scientific reports, discussing and contrasting previously published experiments on the subject. It is a fascinating topic, the conclusions are in concordance with the reviewed material, and the figures and table are adequate.
The review is exhaustive and is formally well thought out and argued with the referenced bibliography, so its publication is desirable.

We appreciate the reviewer’s understanding. No specific issue has been raised.

Reviewer 3 Report

Authors state in their summary:

 While brain disorders emerge from pathological processes at molecular and 10 cellular levels, recent studies have demonstrated that controlling brain activity can modify molecular and cellular pathologies as well as cognitive functions.

However, in their entire analysis and literature survey they have ignored the molecular and cellular changes in both mice and human studies.

Authors make a generalized comment:

For example, abnormalities in gamma oscillations have  been recognized as a neurophysiological marker for various neuropsychiatric disorders and neurodegenerative diseases, such as schizophrenia [33], autism spectrum disorder [34,35], depression [36,37], and Alzheimer’s disease (AD) [30].

They need to define a few of these abnormalities related to  specific  conditions  and elaborate on the factors that promote  such abnormalities or contribute to their delayed onset.  

Authors write again: 

While abnormalities in EEG patterns have long been recognized since Hans Berger [29], 84 accumulating evidence indicates that neuromodulation approaches have a potential to  modify disease states [43,79].

Neuromodulation is accomplished at multiple  levels including diverse pathways of  both limbic and olfactory systems as well as both neuronal and non-neuronal cell types. Authors elaborate on invasive and non-invasive strategies and their pathological outcome. However, there are multiple steps that lead to that particular pathological state that authors have completely ignored.  Below I am listing some articles that would help authors to get insight into this aspect and include these in their review.

  1. Brechet et al. Improving autobiographical memory in Alzheimer’s disease by transcranial alternating current stimulation, Current Opinion in Behavioral Sciences 2021, 40:64–71 doi.org/10.1016/j.cobeha.2021.01.003
  2. Isla et al. Efficacy of preclinical pharmacological interventions against alterations of neuronal network oscillations in Alzheimer's disease: A systematic review. Experimental Neurology 343 (2021) 113743 org/10.1016/j.expneurol.2021.113743
  3. Perez et al. 2020 Protein Biomarkers for the Diagnosis of Alzheimer’s Disease at Different Stages of Neurodegeneration Int. J. Mol. Sci. 2020, 21, 6749; doi:10.3390/ijms21186749
  4. Wang J et al. Enhanced Gamma Activity and Cross-Frequency Interaction of Resting-State Electroencephalographic Oscillations in Patients with Alzheimer’s Disease doi: 10.3389/fnagi.2017.00243

Authors write:

As we discussed above, although reduced gamma power has consistently been reported in mouse models, available evidence in human patients is conflicting [120-127].

Rodents use brain primarily for foraging and learning reproduction related cues. This is in sharp contrast with complicated human brain function. In addition, there are marked differences in  neurogenesis in rodents and humans. These provide clues for differential and conflicting evidence. There are some excellent reviews that authors could look at and come up with an explanation that may be logical and probable.

Authors have identified the cell types and some molecules involved in slow oscillation. They even cite the reference (178), however, they have completely ignored  the regulatory molecule and its probable role. 

Authors have identified the cell types and some molecules involved in slow oscillation. They even cite the reference (178), however, they have completely ignored  the regulatory molecule and its probable role as evident in the interactions analyzed in the article. The same remains true for reference 201. Authors need to include additional information pertaining to respective protein accumulation pathology that is hallmark of AD as well as its link to sleep deprivation. 

Along the same line, further elaboration of the results from references 218 and 223 is necessary.

I think this additional work will be helpful,  will enhance the quality of the manuscript making it more  appropriate for journal and acomprehensive review for the field.

Author Response

Thank you very much for your constructive feedback. Our point-by-point responses are as follows. All changes are highlighted in the revised manuscript.

Authors state in their summary:

 While brain disorders emerge from pathological processes at molecular and 10 cellular levels, recent studies have demonstrated that controlling brain activity can modify molecular and cellular pathologies as well as cognitive functions.

However, in their entire analysis and literature survey they have ignored the molecular and cellular changes in both mice and human studies.

We have rephrased the sentence so that it can reflect what we have primarily reviewed. Please see lines 10-12. 

Authors make a generalized comment:

For example, abnormalities in gamma oscillations have  been recognized as a neurophysiological marker for various neuropsychiatric disorders and neurodegenerative diseases, such as schizophrenia [33], autism spectrum disorder [34,35], depression [36,37], and Alzheimer’s disease (AD) [30].

They need to define a few of these abnormalities related to  specific  conditions  and elaborate on the factors that promote  such abnormalities or contribute to their delayed onset.  

We have added descriptions about the abnormalities in gamma oscillations of these neuropsychiatric disorders and neurodegenerative diseases. Please see lines 56-59. 

Authors write again: 

While abnormalities in EEG patterns have long been recognized since Hans Berger [29], 84 accumulating evidence indicates that neuromodulation approaches have a potential to  modify disease states [43,79].

Neuromodulation is accomplished at multiple  levels including diverse pathways of  both limbic and olfactory systems as well as both neuronal and non-neuronal cell types. Authors elaborate on invasive and non-invasive strategies and their pathological outcome. However, there are multiple steps that lead to that particular pathological state that authors have completely ignored.  Below I am listing some articles that would help authors to get insight into this aspect and include these in their review. 

  1. Brechet et al. Improving autobiographical memory in Alzheimer’s disease by transcranial alternating current stimulation, Current Opinion in Behavioral Sciences 2021, 40:64–71 doi.org/10.1016/j.cobeha.2021.01.003
  2. Isla et al. Efficacy of preclinical pharmacological interventions against alterations of neuronal network oscillations in Alzheimer's disease: A systematic review. Experimental Neurology 343 (2021) 113743 org/10.1016/j.expneurol.2021.113743
  3. Perez et al. 2020 Protein Biomarkers for the Diagnosis of Alzheimer’s Disease at Different Stages of Neurodegeneration Int. J. Mol. Sci. 2020, 21, 6749; doi:10.3390/ijms21186749
  4. Wang J et al. Enhanced Gamma Activity and Cross-Frequency Interaction of Resting-State Electroencephalographic Oscillations in Patients with Alzheimer’s Disease doi: 10.3389/fnagi.2017.00243

Thank you very much for the suggestion. We have made the following amendments by citing all of the suggested articles:

Firstly, article#1 has been cited as Ref 67 in lines 74 and 105; article# 2 has been cited as Ref 93 in line 105; article#3 has been cited as Ref 85 in line 86. 

Secondly, article#4 has been cited as Ref 133 and discussed in lines 175-178.

Authors write:

As we discussed above, although reduced gamma power has consistently been reported in mouse models, available evidence in human patients is conflicting [120-127].

Rodents use brain primarily for foraging and learning reproduction related cues. This is in sharp contrast with complicated human brain function. In addition, there are marked differences in  neurogenesis in rodents and humans. These provide clues for differential and conflicting evidence. There are some excellent reviews that authors could look at and come up with an explanation that may be logical and probable.

We have elaborated our discussion on this issue. Please see lines 234-248 and 453-460.

Authors have identified the cell types and some molecules involved in slow oscillation. They even cite the reference (178), however, they have completely ignored  the regulatory molecule and its probable role as evident in the interactions analyzed in the article.

We have added brief descriptions about the molecular mechanism of slow oscillations. Please see lines 339-344.

The same remains true for reference 201.

Regarding the molecular change by sleep, we have referred to recent papers which have comprehensively characterised transcriptomic and synaptic phosphorylation profiles related to sleep-wake cycles. Please see lines 319-322 and Refs 185 and 186.

Authors need to include additional information pertaining to respective protein accumulation pathology that is hallmark of AD as well as its link to sleep deprivation. 

We have added information related to this point. Please see lines 303-307 and Refs 179 and 180.

Along the same line, further elaboration of the results from references 218 and 223 is necessary.

Regarding the Ref 218 (Holth et al., 2017), now 237, we have added a new table to summarise all referred studies. Please see Table 3.

Regarding the Ref 223 (Rodriguez et al., 2020), now 245, we have discussed this study. Please see lines 477-481.

I think this additional work will be helpful,  will enhance the quality of the manuscript making it more  appropriate for journal and acomprehensive review for the field.

Again, thank you for all the constructive comments.

Reviewer 4 Report

I am reviewing the original draft of the manuscript entitled "
 Mutual interactions between brain states and Alzheimer’s disease pathology "

 In this review, the authors provide an overview of brain activity in Alzheimer's disease. They summarise how brain activity changes in humans and mouse models and how different strategies for neuromodulation can modify AD pathology. This is a very interesting and emerging field and can provide a useful tool for non-pharmacological treatment. 

I only have some comments and suggestions for the Authors. I am sure that the changes suggested can be successfully covered.

- Title: following the reading of this interesting review, I do not find that the title matches well with the content. The title should be more explanative and focusing on slow and gamma oscillations specifically. Brain states involve a very much wide field than the one covered in the review. 

- Line 66. In their first reference to DBS, the authors could briefly explain this technique to make it clear that it is invasive, in contrast to TMS.

- To better explain the point of the authors, I also miss a section about how gamma and slow oscillations contribute to normal cognitive processing. 

- The authors focalise their study on gamma and slow oscillations. I believe that this choice is appropriate. However, in the Introduction, it would be helpful an overview of oscillatory changes in AD since a general slowing is usually described, and theta-gamma interactions are also a hallmark. The authors should justify a little more their focus in the review. 

- Although the recognition of his seminal work is very relevant, Hans Berger is a little bit over cited. It seems repetitive. 

I also suggest including in the Introduction the notion that the frequency of brain oscillations is linked to their spatial distribution, being fast frequency oscillations more related to local neuronal spiking and slower waves capable of integrating larger and more distributed neuronal networks populations. I especially like the description in Buzsáki et al., 2013 Scaling Brain Size, Keeping Timing: Evolutionary Preservation of Brain Rhythms.

- The authors include relevant tables summarizing the studies about neuromodulation. Similar tables, including the mentioned studies on the alterations in gamma and slow oscillations and their findings, are highly recommended. 

- "Gamma oscillations and AD in humans" section: I am missing a mention about whether the literature reflects differences in gamma activity across cortical areas. Do the findings refer to the temporal lobe, prefrontal cortex or others? 

- Is there any difference in gamma activity between MCI and AD patients?

- Figure 3. A- should be included in the labelling. 

- "Gamma oscillations and AD in mouse models" section:

The authors state: "abnormalities in the coupling of gamma oscillations with sharp wave-ripples or theta oscillations have been consistently observed". Which abnormalities? The text should be more specific here. For example, Zhang et al., 2016 found a reduction in theta-gamma  coupling without reduced gamma power. These findings contrast to the sentence "it is clear that reduction in gamma power is associated with 204 AD pathology, at least in mouse models ". The review should include both points of view to be more objective. 

- Have the authors found any reference to the effect of amyloid or tau over hippocampal or cortical interneurons?

- The authors cite interesting studies of neuromodulation of gamma oscillations. However, they don't indicate whether these studies were conducted in rodent models or patients. For optogenetic studies, the answer is clear, but multisensory stimulation can lead to misunderstanding. 

- Figure 4. Note that the type is changed in the legend. The computational models are nice and explanative. 

- Also, the glymphatic system is more active in removing waste solutes during NREM sleep. This link has also been explored for its importance in neurodegenerative disorders, including AD. This point is mentioned in the Conclusions, but I think it should better to explain it in the Slow waves section.

Author Response

I am reviewing the original draft of the manuscript entitled "
 Mutual interactions between brain states and Alzheimer’s disease pathology "

 In this review, the authors provide an overview of brain activity in Alzheimer's disease. They summarise how brain activity changes in humans and mouse models and how different strategies for neuromodulation can modify AD pathology. This is a very interesting and emerging field and can provide a useful tool for non-pharmacological treatment. 

Thank you very much for constructive feedback. Our point-by-point responses are as follows.

I only have some comments and suggestions for the Authors. I am sure that the changes suggested can be successfully covered.

- Title: following the reading of this interesting review, I do not find that the title matches well with the content. The title should be more explanative and focusing on slow and gamma oscillations specifically. Brain states involve a very much wide field than the one covered in the review. 

Agree. We have updated the title as “Mutual interactions between brain states and Alzheimer’s dis-ease pathology: a focus on gamma and slow oscillations”.

- Line 66. In their first reference to DBS, the authors could briefly explain this technique to make it clear that it is invasive, in contrast to TMS.

We have briefly explained what DBS is. Please see lines 67-69.

- To better explain the point of the authors, I also miss a section about how gamma and slow oscillations contribute to normal cognitive processing. 

Regarding gamma oscillations, please see lines 41-44.

Regarding slow oscillations, please see lines 48-50.

In addition to these brief introductory sentences, each starting paragraph in the relevant section also describes the function of these oscillations.

- The authors focalise their study on gamma and slow oscillations. I believe that this choice is appropriate. However, in the Introduction, it would be helpful an overview of oscillatory changes in AD since a general slowing is usually described, and theta-gamma interactions are also a hallmark. The authors should justify a little more their focus in the review. 

We have provided a brief overview of this topic and have elaborated the justification of our focus on gamma and slow oscillations. Please see lines 89-99.

- Although the recognition of his seminal work is very relevant, Hans Berger is a little bit over cited. It seems repetitive. 

Agree. We have made the following amendments:

Firstly, we have rephrased the original one. Please see line 89.

Secondly, we have omitted the repetitive sentence. Please see line 107.

I also suggest including in the Introduction the notion that the frequency of brain oscillations is linked to their spatial distribution, being fast frequency oscillations more related to local neuronal spiking and slower waves capable of integrating larger and more distributed neuronal networks populations. I especially like the description in Buzsáki et al., 2013 Scaling Brain Size, Keeping Timing: Evolutionary Preservation of Brain Rhythms.

Thank you for the suggestion. We have updated the descriptions about various neural oscillations by referring to Buzsaki et al. (2013) as Ref 3. Please see lines 38 and 41-50.

- The authors include relevant tables summarizing the studies about neuromodulation. Similar tables, including the mentioned studies on the alterations in gamma and slow oscillations and their findings, are highly recommended. 

We have compiled all cited studies. Please see new Tables 1 and 3.

- "Gamma oscillations and AD in humans" section: I am missing a mention about whether the literature reflects differences in gamma activity across cortical areas. Do the findings refer to the temporal lobe, prefrontal cortex or others? 

We have elaborated our descriptions about varied experimental conditions. Please see lines 175-183.

- Is there any difference in gamma activity between MCI and AD patients?

We have added comments on this. Please see lines 179-181.

- Figure 3. A- should be included in the labelling. 

We have clarified all labels. Please see the caption in Figure 3.

- "Gamma oscillations and AD in mouse models" section:

The authors state: "abnormalities in the coupling of gamma oscillations with sharp wave-ripples or theta oscillations have been consistently observed". Which abnormalities? The text should be more specific here. For example, Zhang et al., 2016 found a reduction in theta-gamma  coupling without reduced gamma power. These findings contrast to the sentence "it is clear that reduction in gamma power is associated with 204 AD pathology, at least in mouse models ". The review should include both points of view to be more objective. 

Thank you for reminding us about the important study by Zhang et al. (2016). We have commented on this study in lines 210-214 and Ref 147. We have interpreted the results from this study carefully because APP knockout is expected to reduce the production of amyloid-beta, rather than the overexpression of it, which is commonly supposed to model Alzheimer’s disease. Thus, we believe that the amendment has taken an objective approach.

- Have the authors found any reference to the effect of amyloid or tau over hippocampal or cortical interneurons?

We have addressed this crucial issue in lines 227-233.

- The authors cite interesting studies of neuromodulation of gamma oscillations. However, they don't indicate whether these studies were conducted in rodent models or patients. For optogenetic studies, the answer is clear, but multisensory stimulation can lead to misunderstanding. 

We have clarified our descriptions. Please see lines 260-261.

- Figure 4. Note that the type is changed in the legend. The computational models are nice and explanative. 

Corrected.

- Also, the glymphatic system is more active in removing waste solutes during NREM sleep. This link has also been explored for its importance in neurodegenerative disorders, including AD. This point is mentioned in the Conclusions, but I think it should better to explain it in the Slow waves section.

Thank you for the suggestion. We have elaborated our discussion on this important topic. Please see lines 307-312 and 433-435.

Round 2

Reviewer 3 Report

There are some typing errors, editors and author could  notice in proofreading.